# The anatomy of transcriptionally active chromatin loops in *Drosophila* primary spermatocytes using super-resolution microscopy

**Madeleine L. Ball**[1☯], **Stefan A. Koestler**[1☯], **Leila Muresan**[2], **Sohaib Abdul Rehman**[2¤], **Kevin O'Holleran**[2], **Robert White**[1] *

1 Department of Physiology, Development and Neuroscience, University of Cambridge, Downing Site, Cambridge, United Kingdom, 2 Cambridge Advanced Imaging Centre, University of Cambridge, Downing Site, Cambridge, United Kingdom

☯ These authors contributed equally to this work.
¤ Current address: Department of Molecular and Cellular Biology, Harvard University, Cambridge, Massachusetts, United States of America
* rw108@cam.ac.uk

**Data Availability Statement:** The STORM localisation data for this paper is available in the shareloc.xyz repository at https://zenodo.org/

## Abstract

While the biochemistry of gene transcription has been well studied, our understanding of how this process is organised in 3D within the intact nucleus is less well understood. Here we investigate the structure of actively transcribed chromatin and the architecture of its interaction with active RNA polymerase. For this analysis, we have used super-resolution microscopy to image the *Drosophila melanogaster* Y loops which represent huge, several megabases long, single transcription units. The Y loops provide a particularly amenable model system for transcriptionally active chromatin. We find that, although these transcribed loops are decondensed they are not organised as extended 10nm fibres, but rather they largely consist of chains of nucleosome clusters. The average width of each cluster is around 50nm. We find that foci of active RNA polymerase are generally located off the main fibre axis on the periphery of the nucleosome clusters. Foci of RNA polymerase and nascent transcripts are distributed around the Y loops rather than being clustered in individual transcription factories. However, as the RNA polymerase foci are considerably less prevalent than the nucleosome clusters, the organisation of this active chromatin into chains of nucleosome clusters is unlikely to be determined by the activity of the polymerases transcribing the Y loops. These results provide a foundation for understanding the topological relationship between chromatin and the process of gene transcription.

## Author summary

Transcription is a highly regulated process enabling the production of different cell types during development and the ability to change gene expression in response to external signals. When transcription regulation goes wrong it can lead to a variety of disease states

record/7501960#.Y-IO-hPP3eo. The DOI is: 10.5281/zenodo.7501960.

**Funding:** The work was supported by the Biotechnology and Biological Sciences Research Council (BBSRC) Grant (BB/S00758X/1) and an Isaac Newton Trust Research (INT) Grant to RW. SAK was supported by the BBSRC and INT grants. MLB was supported by a Cambridge BBSRC-DTP studentship, SAR was supported by the Cambridge Commonwealth, European and International Trust and the Higher Education Commission, Pakistan and LM was supported by an Engineering and Physical Sciences Research Council Grant (EP/R025398/1). The funders had no role in study design, data collection and analysis, decision to publish, or preparation of the manuscript.

**Competing interests:** The authors declare that they have no conflict of interest.

including cancer. In eukaryotes, transcription operates on chromatin, a polymer of repeating nucleosomes formed by packaging DNA around histone proteins. Chromatin organisation is linked to transcription regulation, as it is thought to control the access of transcription machinery to the DNA. However, directly imaging chromatin within the nucleus is traditionally very difficult due to the dense packaging. *Drosophila* primary spermatocytes have large nuclei with easily visualised transcriptionally active regions of the Y chromosome known as 'Y loops'. Using super-resolution imaging on Y loops, we show that this active chromatin has a specific organisation as a chain of nucleosome clusters. We also find that actively elongating RNA Polymerase is considerably less prevalent along the loops than the clusters. We conclude that the organisation of active chromatin into nucleosome clusters is unlikely to be caused by transcription elongation. This finding is an important step towards understanding the complex topological relationship between chromatin structure and regulation of transcription.

## Introduction

Considerable advances have been made recently in understanding the organisation of chromatin in the nucleus and its relevance for the function and regulation of the genome (for a recent review see [1]). Genomic studies have provided insights into the variety of levels of organisation ranging from chromatin loops [2,3] and topologically associated domains [4,5], to spatially segregated compartments of condensed repressed chromatin and decondensed active chromatin [6] and chromosome territories [7]. Imaging approaches have provided a complementary view, revealing a dynamic landscape of chromatin structures associated with transcription and replication [8,9]. A fundamental insight emerged from EM studies indicating that chromatin forms a disordered chain within the interphase nucleus with a variety of local structural motifs based on configurations of the 10nm nucleosomal fibre [10]. Other studies have revealed that chromatin is locally organised into small nucleosomal clusters of various sizes (variously termed clutches or domains [11,12]) and super-resolution microscopy has begun to define the structural characteristics of different chromatin states [13,14]. However, linking specific chromatin configurations to the different activities carried out by chromatin remains an important objective if we are to understand how the structure of chromatin facilitates and regulates its many functions. For example, transcriptional activation has long been associated with chromatin decondensation [15–19] but we know little of the detailed chromatin environment encountered by an elongating polymerase.

Imaging chromatin structure within the nucleus is, however, challenging as in most nuclei chromatin is densely packed. Here we take advantage of the exceptionally large nuclei of *Drosophila* primary spermatocytes which show an ordered arrangement of separate chromosomes and a large nucleoplasmic space [20,21]. Much of the nucleoplasm is engaged in the transcription of just a few genes on the Y-chromosome, the enormous genes of the Y loops. These genes, which are specifically activated in the primary spermatocytes, provide an attractive model for the study of transcriptionally active chromatin. The three genes, kl-5, kl-3 and ks-1, decondense upon activation and extend as Y loops in the nucleoplasm [20–22]. These Y loop genes are transcribed as single transcription units, several megabases in length [23,24]. For example, one of these Y loop genes, kl-3 which encodes an axonemal dynein heavy chain spans at least 4.3Mb, although most of this sequence is intronic and its coding sequence is only ~14kb [20,22–24]. These Y loops, easily visible in phase-contrast light microscopy, have long been recognised as models for the organisation of transcriptionally active chromatin providing

a paradigm for gene activation, chromatin decondensation, loop formation and the topology of RNA polymerase progression [15].

In this study, we have focused on the use of super-resolution microscopy to investigate the chromatin structure of Y loops and its relationship to transcription. We find that the chromatin fibre in these transcriptionally active loops is not simply extended as a 10nm fibre but rather is largely organised as a chain of nucleosomal clusters. RNA polymerase is associated with the periphery of these clusters. Comparison of the nucleosome cluster versus RNA polymerase prevalence suggests that the chromatin clustering is not generated by RNA polymerase activity between the clusters. It appears that the chain of nucleosomal clusters forms the basic structure of decondensed chromatin upon which RNA polymerase acts.

## Results

### The fine structure of Y loop chromatin

In *Drosophila* spermatogenesis, after the last spermatogonial mitosis, the primary spermatocytes enter a G2 phase that lasts for several days. In this period they expand over 20-fold in volume and activate the spermatogenesis transcription program. As part of this program, three individual Y chromosome genes decondense from the Y chromosome mass, which is located close to the nucleolus, and spill out into the nucleoplasm forming the Y loops. These actively transcribed loops can be easily visualised by DNA labelling or anti-histone immunolabelling and in confocal microscopy, the loops appear as chromatin ribbons that follow convoluted paths within the nucleus (Fig 1A and 1B). Close examination indicates that these ribbons may not have a uniform structure and often they exhibit a "chain of blobs" appearance. To investigate the structure of this transcriptionally active chromatin in more detail we have examined the loops in super-resolution using Stochastic Optical Reconstruction Microscopy (STORM) [25].

STORM imaging of Y loop chromatin labelled with anti-histone antibody reveals the fine structure of the Y loops predominantly as chains of small chromatin clusters (Fig 1C). At higher magnification (Fig 2) sparse localisations can be seen in the "links" between the clusters indicating that the linking chromatin is also nucleosomal. Using a series of z-slices to sample a z-depth of over 1.5μm demonstrates that the clustered appearance of the Y loop chromatin represents genuine 3D clusters and is not due to optical sectioning of a restricted focal depth (Fig 2).

To model the spatial aggregation of antibody labelled nucleosomes within the Y loop clusters, several clustering methods were considered to identify and describe the variability in cluster sizes; including DBSCAN [26], a Bayesian approach [27] and the mode-finding clustering algorithm MeanShift [28–30]. We found MeanShift to be more robust than the alternatives [26,27,31] with respect to identifying clusters closely positioned along a fibre. The crucial input parameter of MeanShift is the search radius. In order to estimate the search radius an average cluster model was identified from the data based on spatial statistics modelling (see Materials and Methods for details), similar to approaches used previously [31–33]. Using this spatial statistics and MeanShift approach to estimate cluster size, the median cluster width (Full Width Half Maximum, FWHM) is 52nm, with an interquartile range from 44nm to 61nm (Fig 3).

The median distance between cluster centres is 107nm (Fig 4). On a simple volume calculation, a sphere of diameter 52nm could accommodate a maximum of 158 nucleosomes, however the density of nucleosomes in the clusters is likely to be much less than this. From EM studies, the chromatin volume concentration (CVC) ranges from 12 to 52% in interphase chromatin, with a mean of 30%, and in heterochromatin the range is 37 to 52% [10]. It is not

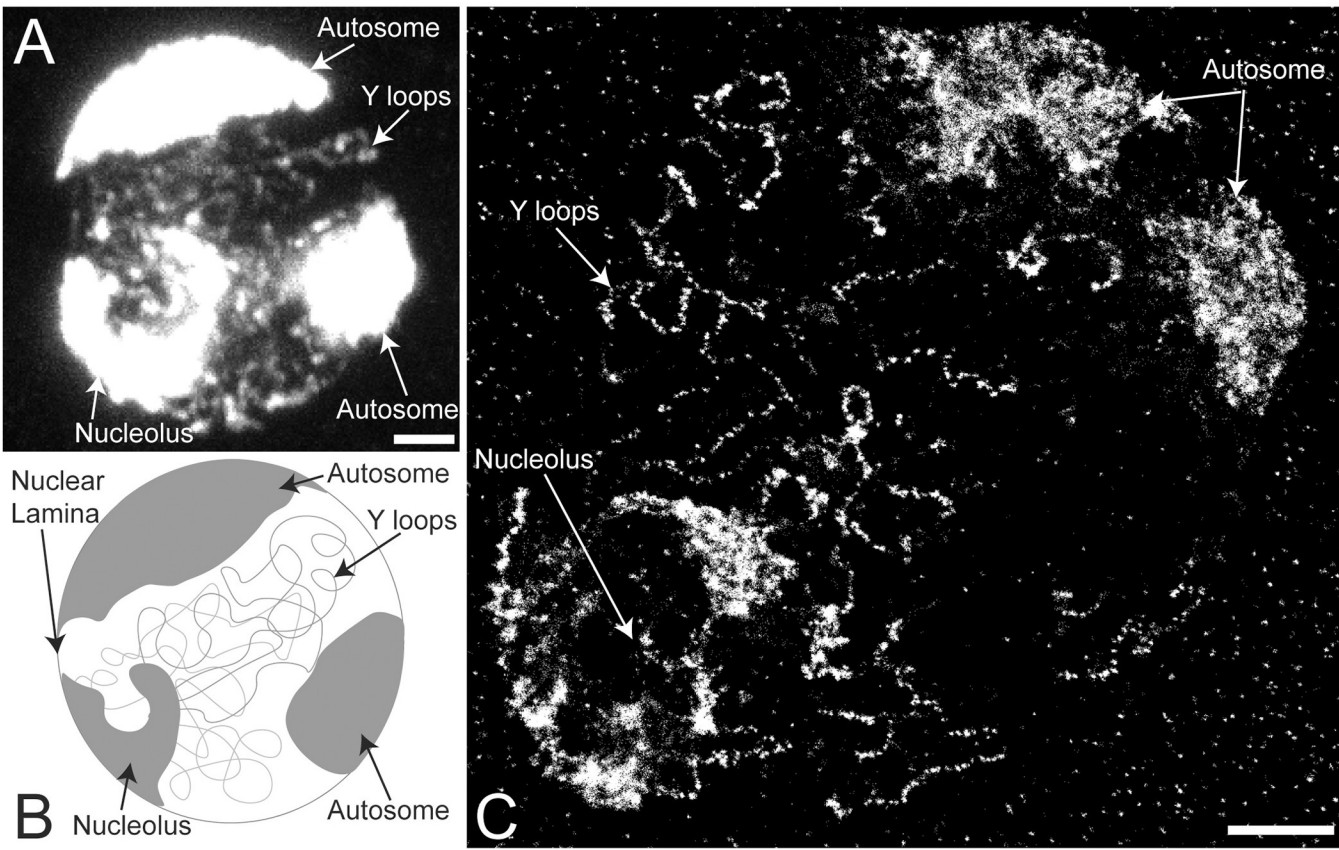

**Fig 1. Overview of the *Drosophila* primary spermatocyte nucleus.** (A) Confocal image of *Drosophila* primary spermatocyte nucleus immuno-labelled with pan-histone antibody labelling core histones plus H1. (B) Schematic showing the characteristic features of the *Drosophila* primary spermatocyte nuclei. (C) STORM super-resolution image of *Drosophila* primary spermatocyte nucleus, immunolabelled with pan-histone antibody. The Y loops, nucleolus, and two lobes of the same autosome are indicated with arrows. Scale bars are 2 μm.

clear what CVC is appropriate for the Y loop clusters but, to provide an estimate of nucleosome number, taking a CVC value of 35% (above the mean but below the value for heterochromatin) would give a median of 54 nucleosomes per cluster (see Materials and Methods for details). The overall suggested structure is schematically diagrammed in Fig 4.

We note that although much of the Y loop chromatin adopts this "chain of clusters" structure, the Y loops do not simply have a uniform structure and, in addition to the individual elongated "chain of clusters" fibres, we also find regions of more aggregated chromatin (Figs 1 and 2).

## Topology of active transcription

The observation that actively transcribed chromatin adopts a cluster chain organisation raises the question of the relationship of the clusters to active RNA polymerase. To examine this we used two-colour STORM imaging with immunolabelling for RNA polymerase II with the phospho-Ser2 modification on the C-terminal domain repeats (RPol-PSer2) representing actively elongating RNA polymerase, together with anti-histone labelling for chromatin. We anticipated that elongating RNA polymerase might occupy the linking regions between the Y loop chromatin clusters however the labelling was surprising in two ways; first, the RPol-PSer2 labelling was sparse relative to the occurrence of cluster linking regions and second, the RPol-

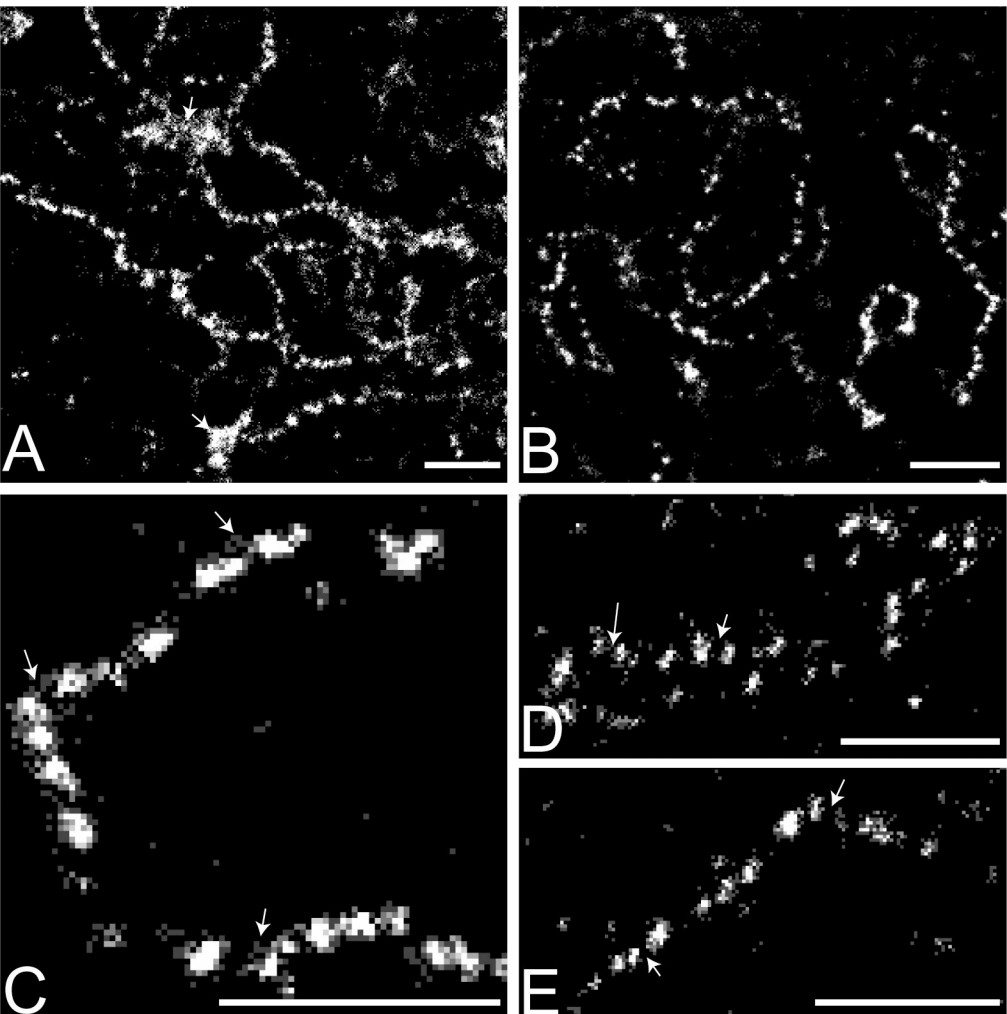

**Fig 2. STORM super-resolution images of the Y loops.** Histones are immunolabelled with pan-histone antibody, showing the clusters of nucleosomes. (A-C) Three representative examples of single slice images from different cells are shown increasing in magnification from A to C. The Y loops are largely made up of semi-regular clusters of nucleosomes. In A there are also some larger aggregates visible, indicated with arrows. In C the arrows indicate evidence of smaller looping regions of chromatin fibres between and extending from the nucleosome clusters. (D and E) Maximum projection images over 1.5 μm of Y loop fibres showing that the clusters are genuine 3D objects as the regions in between clusters of nucleosomes do not fill in and form a complete contiguous fibre; examples indicated with arrows. Scale bars are 1 μm.

PSer2 labelling was not on the main chromatin fibre axis, rather the RPol-PSer2 localisation was peripherally associated with chromatin clusters (Fig 5).

The first observation suggests that the cluster chain organisation is not dictated by the distribution of active polymerase and the second indicates a specific novel topological relationship between the cluster chain and the organisation of foci of active transcription. We confirmed that the observed displacement of the RPol-PSer2 localisations from the fibre axis was not due to camera misalignment (see S1 Information). The distance of RPol-PSer2 from the main fibre axis shows quite a broad distribution with an average value of 100nm and the individual RPol-PSer2 foci have a median width of 48nm (Fig 6).

The chromatin clusters are often associated with sparse anti-histone localisations surrounding the dense cluster core; these may represent decondensed loops emanating from the clusters

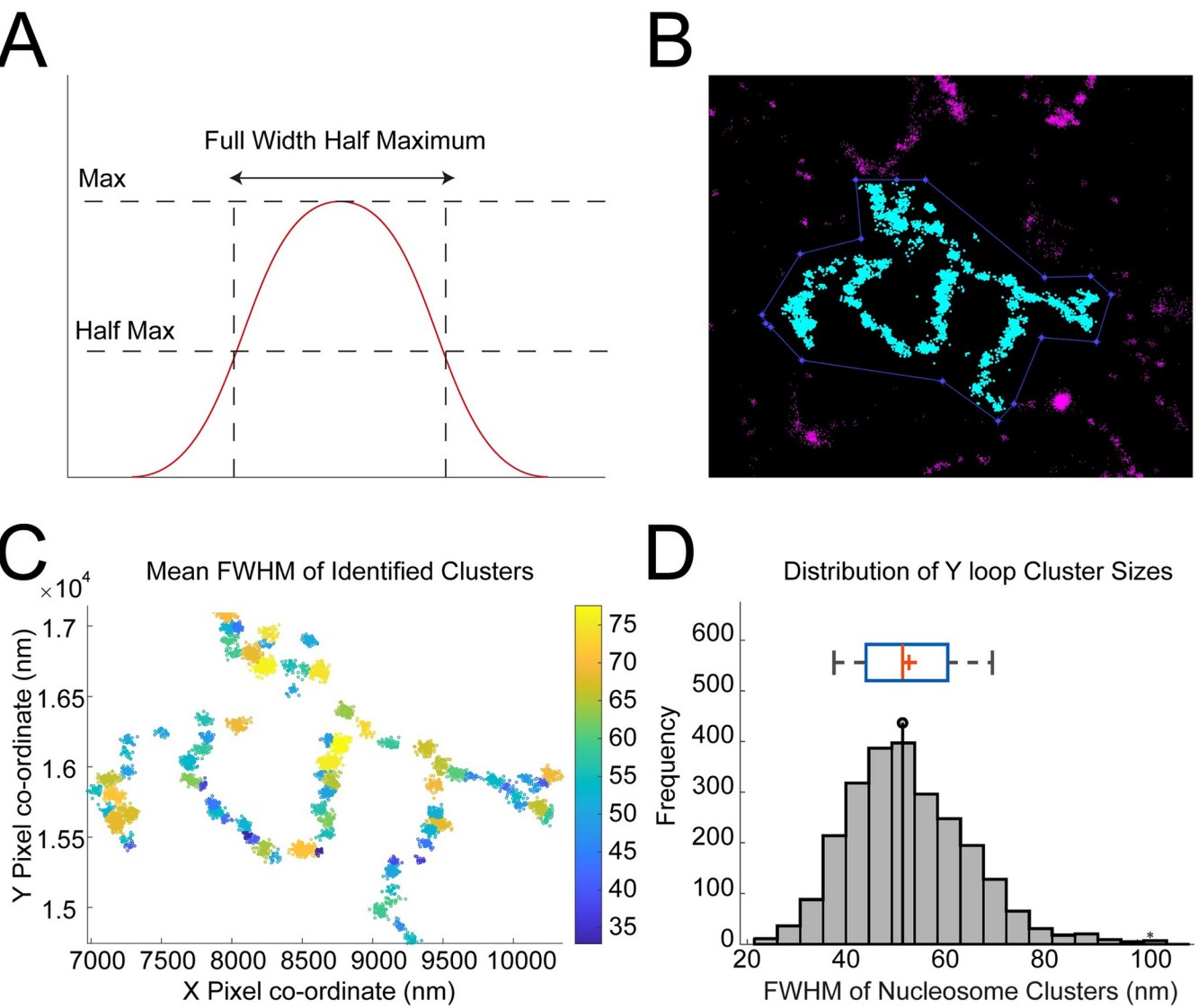

**Fig 3. Quantification of chromatin clusters of the Y loops.** (A) A graphical representation of a theoretical Full Width Half Maximum (FWHM). The width of the distribution is taken at point of half the maximum signal. This is a measure of diameter that can exclude less confident edge points. (B) To quantify the sizes of the clusters along the Y loops, 32 regions of interest (ROI) were selected from 12 cells with the highest resolution, and with many Y loops visible, resulting in a total of 2473 clusters for analysis. An example of a cropped ROI of the Y loop chromatin is indicated in cyan, unselected regions are shown in magenta and the ROI boundary is indicated by the dotted blue line. (C) Pair Correlation Function (PCF) fitting was used to estimate the optimal search radius input, then the data were processed through the MeanShift algorithm to identify clusters. The heatmap scale is in nm. (D) The total distribution of FWHMs of the nucleosome clusters along the Y loops displayed with a histogram, and box and whiskers plot (the box indicates the inter-quartile range and the whiskers show the 9% and 91% bounds). The median FWHM was 52nm, indicated with a lollipop on the histogram, and a line on the box and whiskers plot. The mean was 53nm, indicated by a '+' on the box and whiskers plot.

and active polymerase complexes associated with the periphery of clusters may be engaged with these loop regions (Fig 5). We note that RPol-PSer2 is not only associated with the extended cluster chain regions of the Y loops but is also found on the surface of larger chromatin aggregates.

As the RPol-PSer2 phosphorylated form of RNA polymerase may not represent all polymerase engaged on the Y loop chromatin, we also investigated the localisation of RNA polymerase II with the phospho-Ser5 modification on the C-terminal domain repeats (RPol-

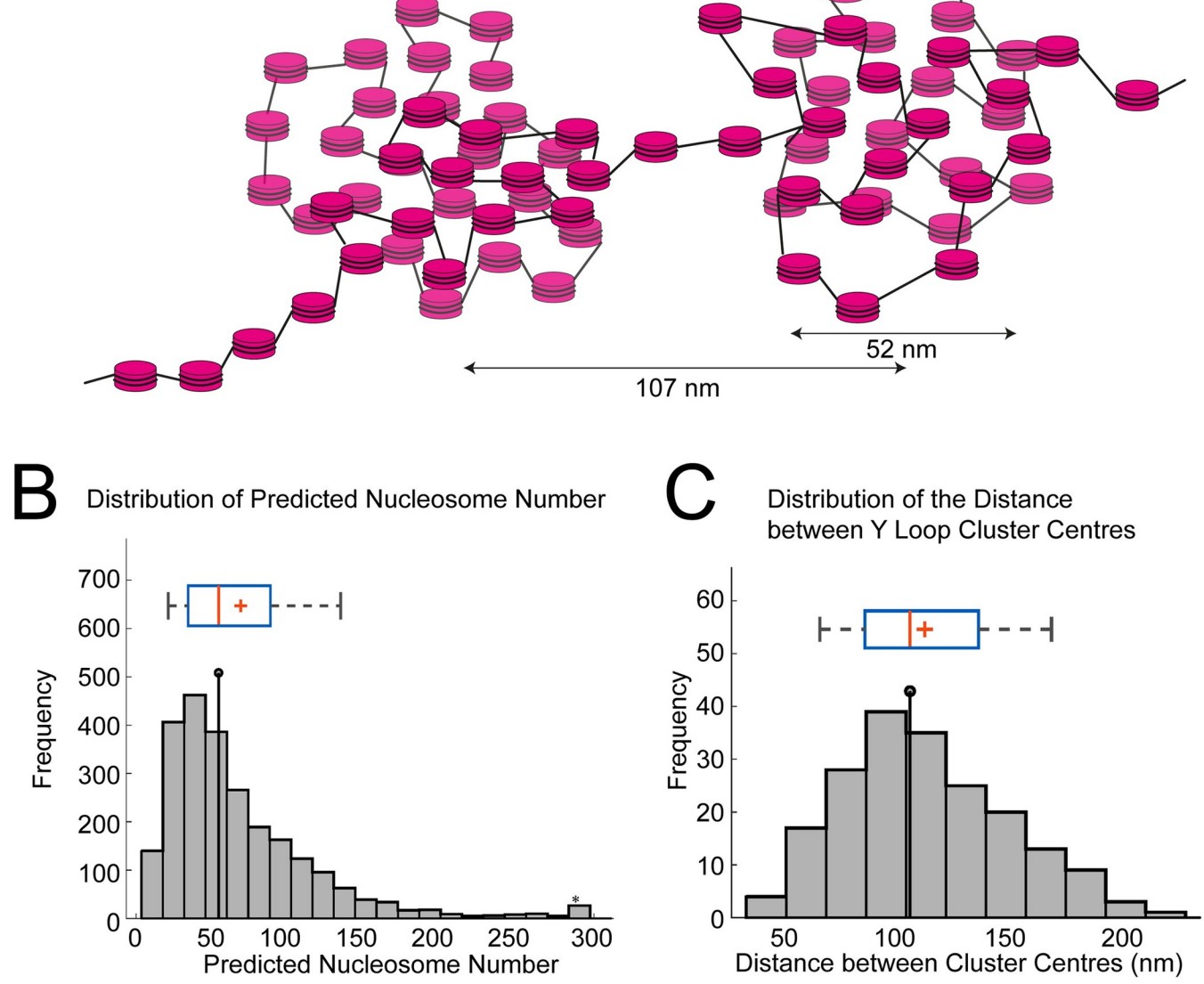

**Fig 4. Estimating the number of nucleosomes in the Y loop clusters.** (A) Schematic of nucleosome clusters along the Y loops with nucleosomes depicted as magenta discs. The median FWHM of the clusters measured was 52nm, shown here against a cluster as an example, as well as the median distance between cluster centres (107nm). (B) Histogram showing the number of predicted nucleosomes within the measured clusters along the Y loops. The FWHM was used to estimate the maximum volume of the clusters and find the theoretical maximum of nucleosomes that would fit within that volume. 35% of this maximum was used as an estimate of nucleosome number based on how condensed the nucleosomes might be within these clusters. The median is 54. (C) Histogram of inter-cluster distance, the median is 107nm. The box and whisker plots are as in Fig 3.

PSer5). The RPol-PSer5 phosphorylated form is associated with the initiation complex, paused polymerase and polymerase interacting with splicing complexes [34,35]. As shown in Fig 7, RPol-PSer5 has a different distribution than RPol-PSer2 and much of the labelling in the nucleoplasm is dispersed and not clearly associated with chromatin fibres.

Some labelling is associated with the Y loops but, as with RPol-PSer2, this is sparse relative to the clusters. We then investigated the combined occurrence of RPol-PSer2 and RPol-PSer5 using an antibody that labels both modified polymerase forms. The distribution of RPol-

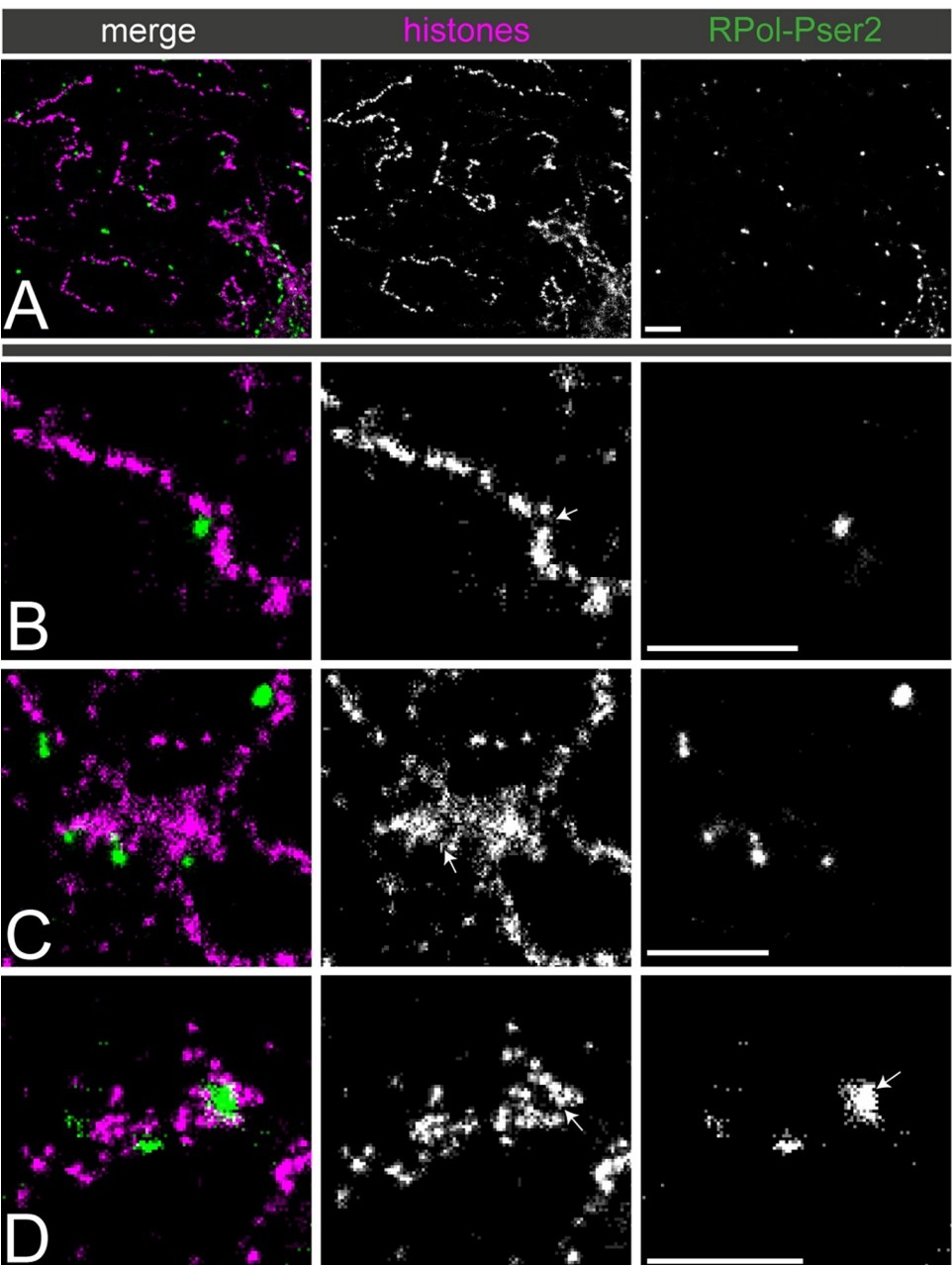

**Fig 5. The association between Y loop chromatin clusters and active polymerase (RPol-PSer2).** Dual-colour STORM images of immunolabelled histones (magenta) and RPol-Pser2 (green). (A) gives an overview showing that polymerase foci are less prevalent than nucleosome clusters. (B to D) show selected examples at higher power. (B) An example of an individual Y loop fibre associated with one isolated focus of RPol-Pser2. RPol II appears to be associated with a looping region inbetween two nucleosome clusters, indicated with an arrow. (C) An example of larger aggregates of chromatin along the Y loops associated with RPol II. The foci of polymerase appear to be associated on the periphery of the aggregates, with some evidence of looping regions, indicated with an arrow. (D) An example of a larger focus of RPol II associated with Y loops. This larger focus could indicate a cluster of polymerase complexes co-transcribing as a "factory". These are rare to see, and do not seem to represent a universal organisation of transcription along the Y loops. Scale bars are 1μm.

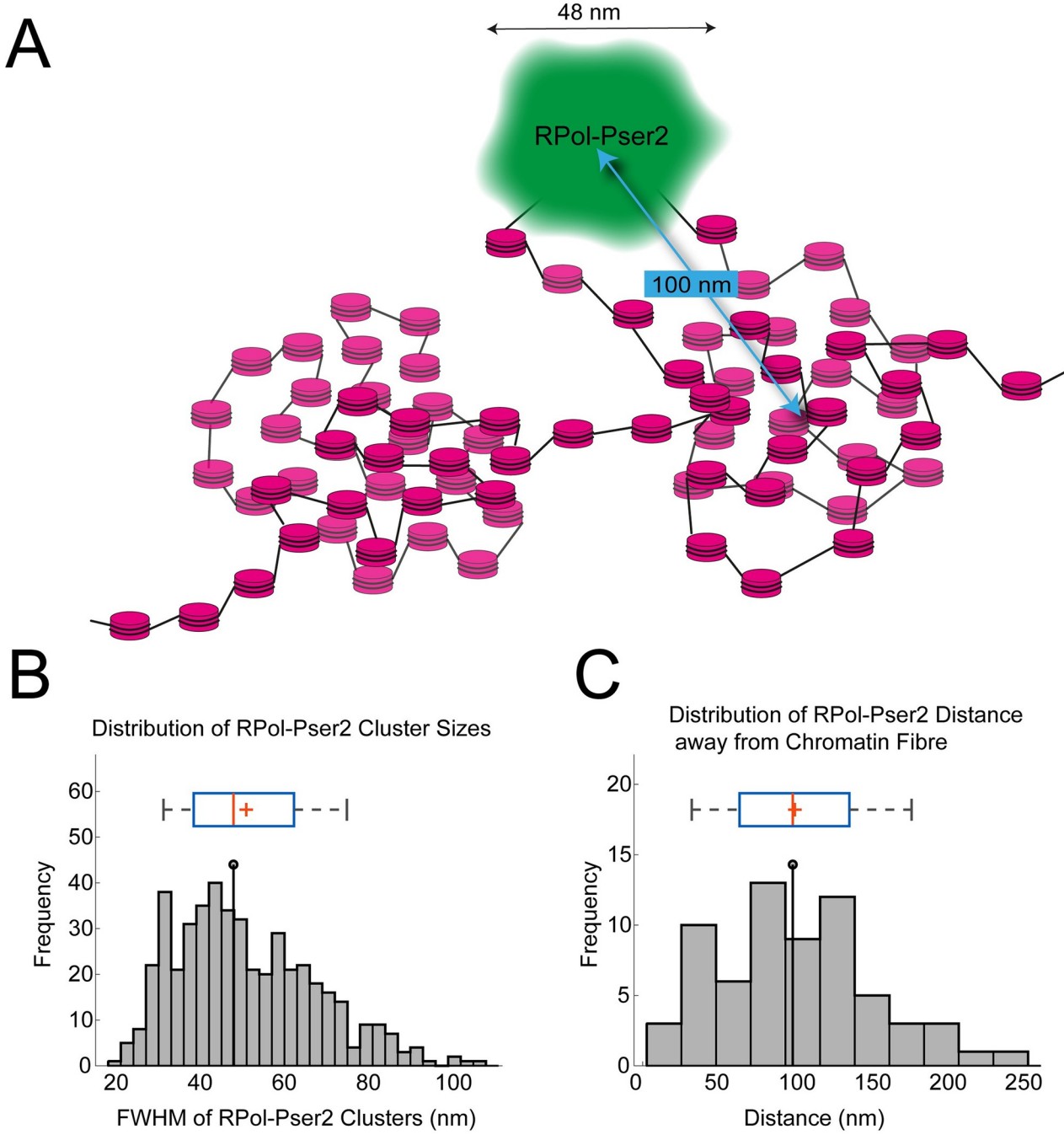

**Fig 6. Quantification of association of Y loops to active transcription.** (A) Schematic showing interpretation of an individual polymerase focus (green) on the periphery of a nucleosome cluster, on a decondensed small loop extending from nucleosome cluster. Arrows indicate average width of polymerase foci and distance from focus centre to centre of nucleosome cluster. (B) Histogram of FWHM of polymerase foci. The median width is 48nm. We note that the distribution appears biphasic or multiphasic suggesting the presence of multiple polymerases in each focus. (C) Histogram of distances between polymerase focus centre to centre of nucleosome cluster, mean is 100nm. The box and whisker plots are as in Fig 3.

PSer2/PSer5 supports the above results (Fig 8), confirming that engaged RNA polymerase does not routinely occur in the cluster linking regions and also that the RNA polymerase localisations are on the periphery of the chromatin clusters and displaced from the main axis of the Y loop chromatin fibres.

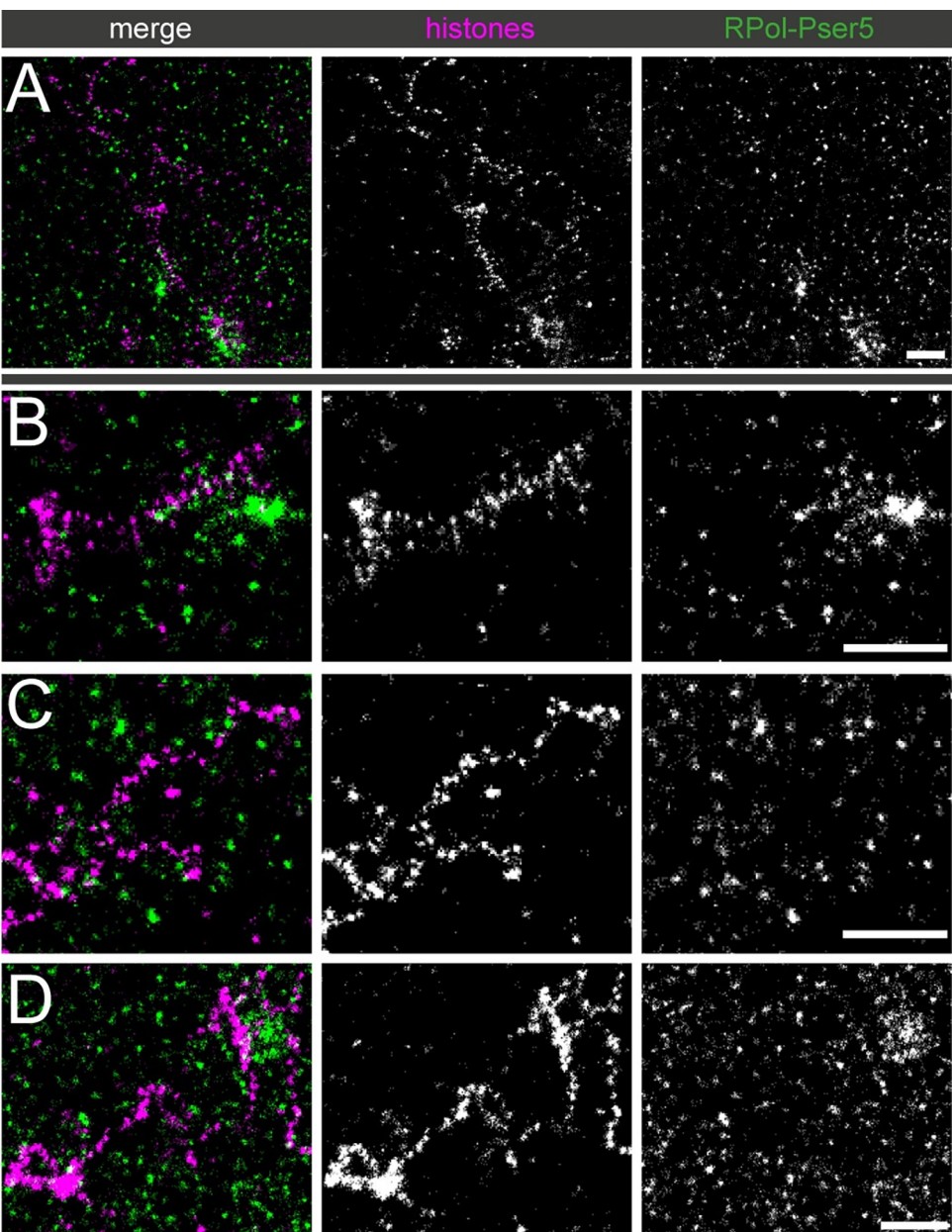

**Fig 7. Association of Y loops to RPol-PSer5.** Dual-colour STORM images of immunolabelled histones (magenta) and RPol-Pser5 (green). (A) gives an overview and (B to D) show selected examples at higher power. Although RPol-PSer5 foci appear to be more prevalent than RPol-PSer2 foci, they are generally not present between the nucleosome clusters. In addition many RPol-PSer5 foci appear distant from chromatin fibres. Scale bars are 1μm.

In order to further assess the role that active transcription plays in the generation of clustered chromatin along the Y loops, transcription inhibition using α-amanitin and triptolide for 20 hours was undertaken. For this we used cultured larval testes to ensure optimal penetration of the drugs into the testes, alongside control larval testes with no drug exposure. The larval testes were used to create a spermatocyte preparation in the same manner as adult testes, the histones were immunolabelled and the nuclei imaged using STORM as before (Fig 9). Despite the transcription inhibition having an effect on the broader phenotype of the nucleus,

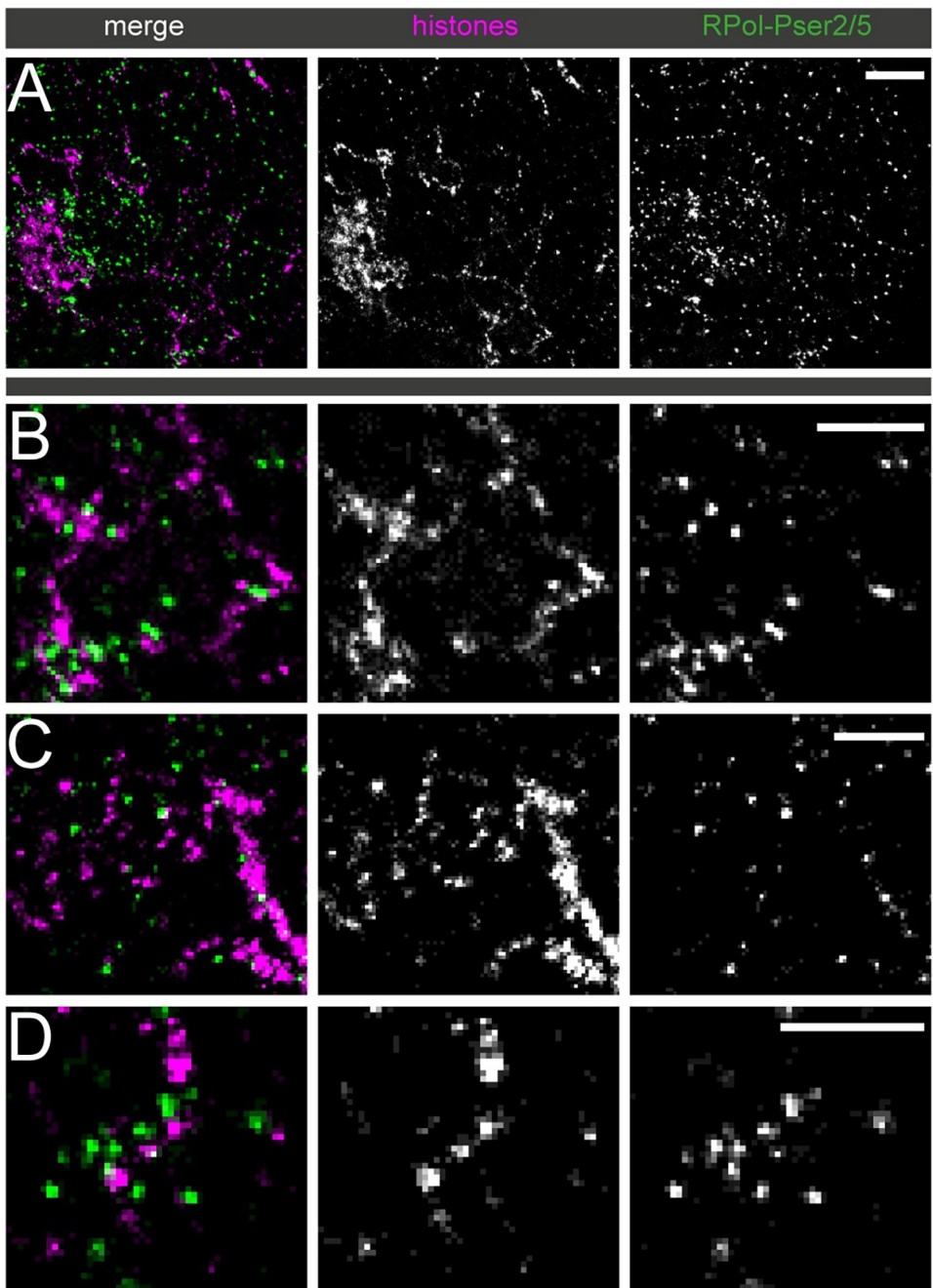

**Fig 8. Association of Y loops to RPol-PSer2/PSer5.** Dual-colour STORM images of immunolabelled histones (magenta) and polymerase foci (green) using an antibody recognizing both RPol-Pser2 and RPol-PSer5. (A) gives an overview, scale bar is 1 μm and (B to D) show selected examples at higher power, scale bars are 500nm. The polymerase foci are generally not present between the nucleosome clusters.

autosomes, and Y loops, the regular pattern of Y loop clusters can still be visualised in the cells (Fig 9D). The larger morphological changes seem to be due to twisting, and collapsing of Y loop clusters towards each other. When comparing ROIs taken from both control and inhibited Y loops overall, there is an increase in average cluster width. However, this is likely due to

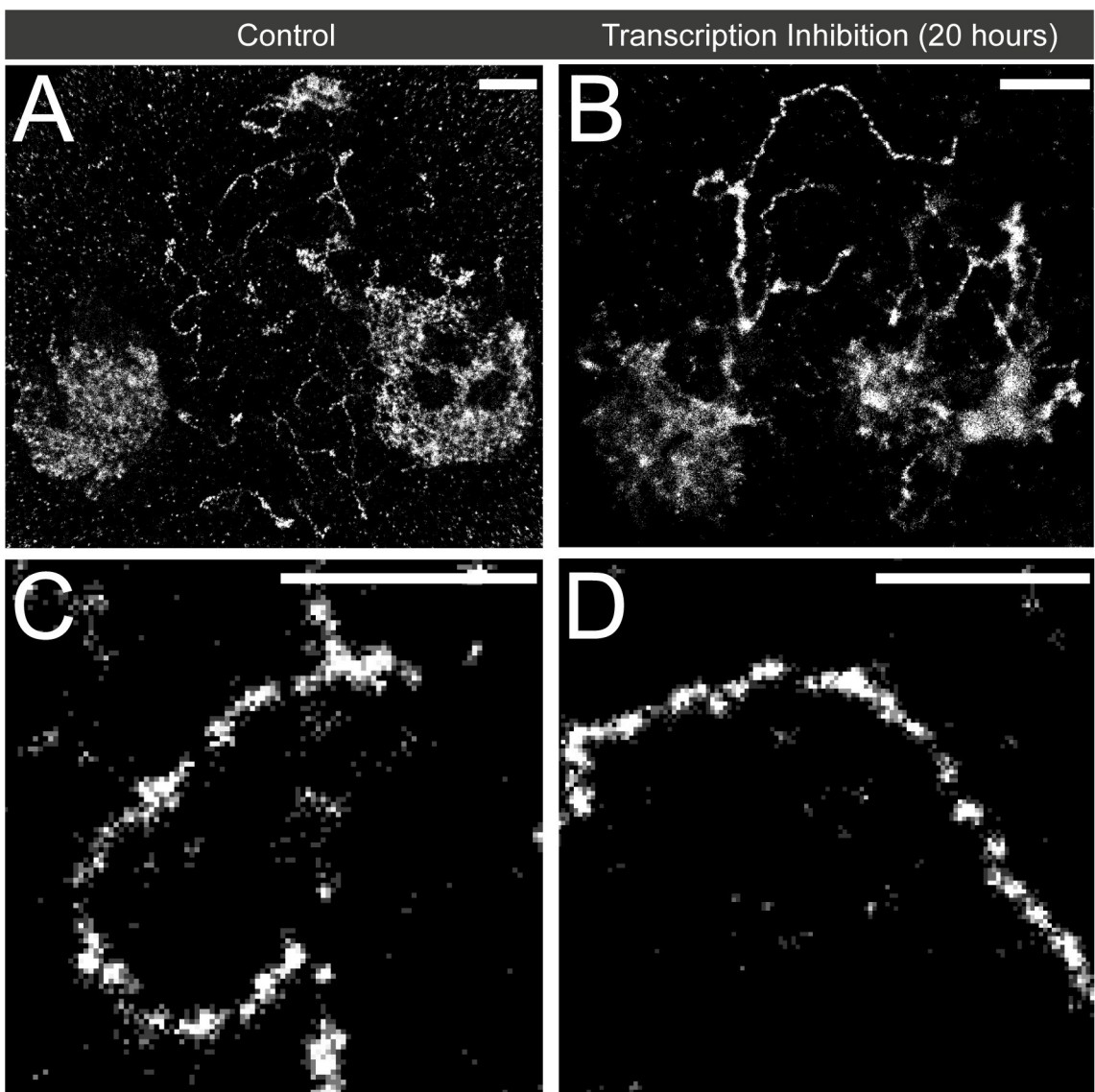

**Fig 9. The effect of transcription inhibition on Y loop chromatin.** (A) An example of a control spermatocyte nucleus with the histone labelled. An autosome is present on the left side of the nucleus, and the nucleolus is on the right side of the nucleus. (B) An example of a spermatocyte nucleus after 20 hours of transcription inhibition using α-amanitin and triptolide. An autosome is also shown on the left side of the nucleus, and the nucleolus is present on the right side of the nucleus. (C) A zoomed in image showing the clusters along the Y loops in the control cell. (D) A zoomed in image showing the clusters along the Y loops in the spermatocytes after transcription inhibition. The scale bars are 2 μm in (A) and (B), the scale bars are 1 μm in (C) and (D).

the presence of broader morphological changes along the Y loops (Fig 9B), as opposed to the regular chromatin clusters becoming larger. To confirm this, more selective ROIs of the inhibited Y loops were taken, removing the presence of larger aggregations for quantification, and these show no difference to the cluster sizes of the control Y loops (see S1 Information for details). Overall, these inhibition experiments support the view that the mechanism forming the local pattern of chromatin clusters is independent of active transcription. The significant inhibition of transcription along the Y loops by our treatment regime was confirmed through EU labelling (see S1 Information).

### Linking RNA polymerase distribution and Y loop transcription

The Y loops represent single transcription units whose large size provides an opportunity to investigate the organisation of nascent transcription in an active transcription loop. We have used the EU-Click-iT assay to image, by laser scanning confocal microscopy (LSCM), the nascent transcription occurring during a short time window (20min) of exposure of the testes to the EU nucleotide. This analysis was undertaken using confocal microscopy, as opposed to using STORM, as the confocal system is more suitable to provide the larger 3D depth needed to understand the distribution of nascent transcription along the Y loops that span almost the entire nucleoplasm. We observe nascent transcription from Y loops as a series of blobs of EU incorporation distributed along extended lengths of the Y loop fibres (Fig 10). This distribution of nascent transcript along the Y loops fits with the distribution of RPol-PSer2, supporting the idea that the RPol-PSer2 localisations observed in the super-resolution images represent actively transcribing polymerase (Fig 5). To further investigate the relationship between nascent transcription and the different modifications on the C-terminal domain of RNA polymerase II we have imaged RPol-PSer2 and RPol-PSer5 in association with EU Click-iT (Figs 10 and 11).

The confocal images of RPol-PSer2 and RPol-PSer5 (Fig 10) provide an overview of the nuclear distribution of the two different modified polymerase forms. RPol-PSer2 is present in puncta on the autosome chromosome masses, on the sex chromosomes (X and Y) associated with the nucleolus and in trails of puncta associated with Y loops in the nucleoplasm. RPol-PSer5 has a similar distribution in puncta on the chromosome masses but has a distinct arrangement as a ring around the nucleolus and more puncta widely dispersed in the nucleoplasm. Imaging RPol-PSer2 together with nascent transcription confirms that RPol-PSer2 is associated with active elongation on the Y loops; as shown in the z-stack in Fig 11A the area of puncta of Y loop nascent transcription is closely associated with strong RPol-PSer2 labelling. In contrast, the Y loop nascent transcription is generally not associated with strong RPol-PSer5 signals and the weak dispersed puncta of RPol-PSer5 labelling in the nucleoplasm do not coincide with active nascent RNA production (Fig 11B). Quantification is provided in the S1 Information showing that the majority of RPol-PSer2 puncta overlap with nascent RNA, whereas for RPol-PSer5 only a minority show overlap. However, close to the nucleolus presumably at the start of a Y loop, a few strong puncta of RPol-PSer5 can be seen extending out from the nucleolus consistent with RPol-PSer5 association with the initiation of transcription. The Y loops thus enable us to visualise the distribution of the modified polymerase forms RPol-PSer2 and RPol-PSer5 along transcription units in vivo within intact nuclei.

## Discussion

We have exploited the large size of the primary spermatocyte nuclei in *Drosophila* as a tractable model system for the super-resolution imaging of chromatin in intact nuclei. We have focussed on the organisation of transcriptionally active chromatin using the Y loop genes which are activated in spermatocytes and extend out from the Y chromosome into the nucleoplasm as huge loops. We find that these active chromatin loops do not simply extend as 10nm "beads-on-a string" nucleosomal fibres but instead have a more complex structure and generally adopt an organisation as chains of nucleosome clusters. We have examined the relationship between this structure and the arrangement of RNA polymerase transcribing the loops and the organisation of nascent transcription.

The chromatin clusters have a relatively tight size distribution with a median width of 52nm and an interquartile range from 44-61nm. To estimate the width we have used the FWHM which provides a robust estimate of width, however it is an underestimate of the full

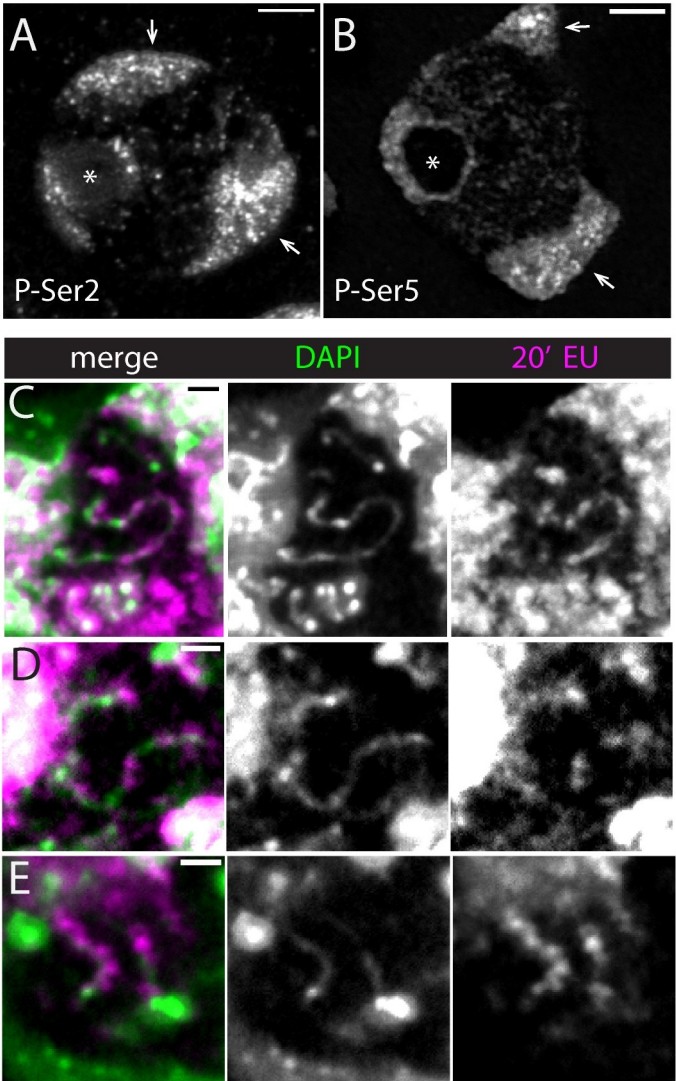

**Fig 10. Polymerase and nascent transcription: confocal overview.** (A and B) Immunofluorescence images of RPol-PSer2 and RPol-PSer5 respectively in primary spermatocytes of larval testis whole-mounts. Single central confocal slices are shown. Arrows indicate autosome masses, asterisk indicates nucleolus, scale bars are 3 μm. (C-E) show nascent RNA along DNA fibres. Nascent RNA labelled by 20min incubation with EU (magenta) gives a blobby appearance along DAPI stained Y loops (green). (C) Maximum intensity projection of five confocal slices with a 150nm step size captures a long Y loop section through the nucleoplasm. (D) and (E) are single slices. Scale bars are 1 μm.

width as the clusters do not have sharp perimeters. The lack of sharpness of the outer edge of clusters may be due to loosening of the compaction at the periphery or the formation of small loops of 10nm fibre extending from the compact clusters. In addition some of the uncertainty of the edge of the clusters will be due to the localisation precision of the STORM and also the labelling method, as the positions of the histones are visualised with an anti-histone primary antibody and a fluorophore-conjugated secondary antibody. Although this estimate is not a direct measurement, it provides a starting point for interpretation of the observed labelling in terms of underlying chromatin structure. As presented above in the Results, a chromatin cluster with a width of 52nm could accommodate 54 nucleosomes using a packing density (CVC

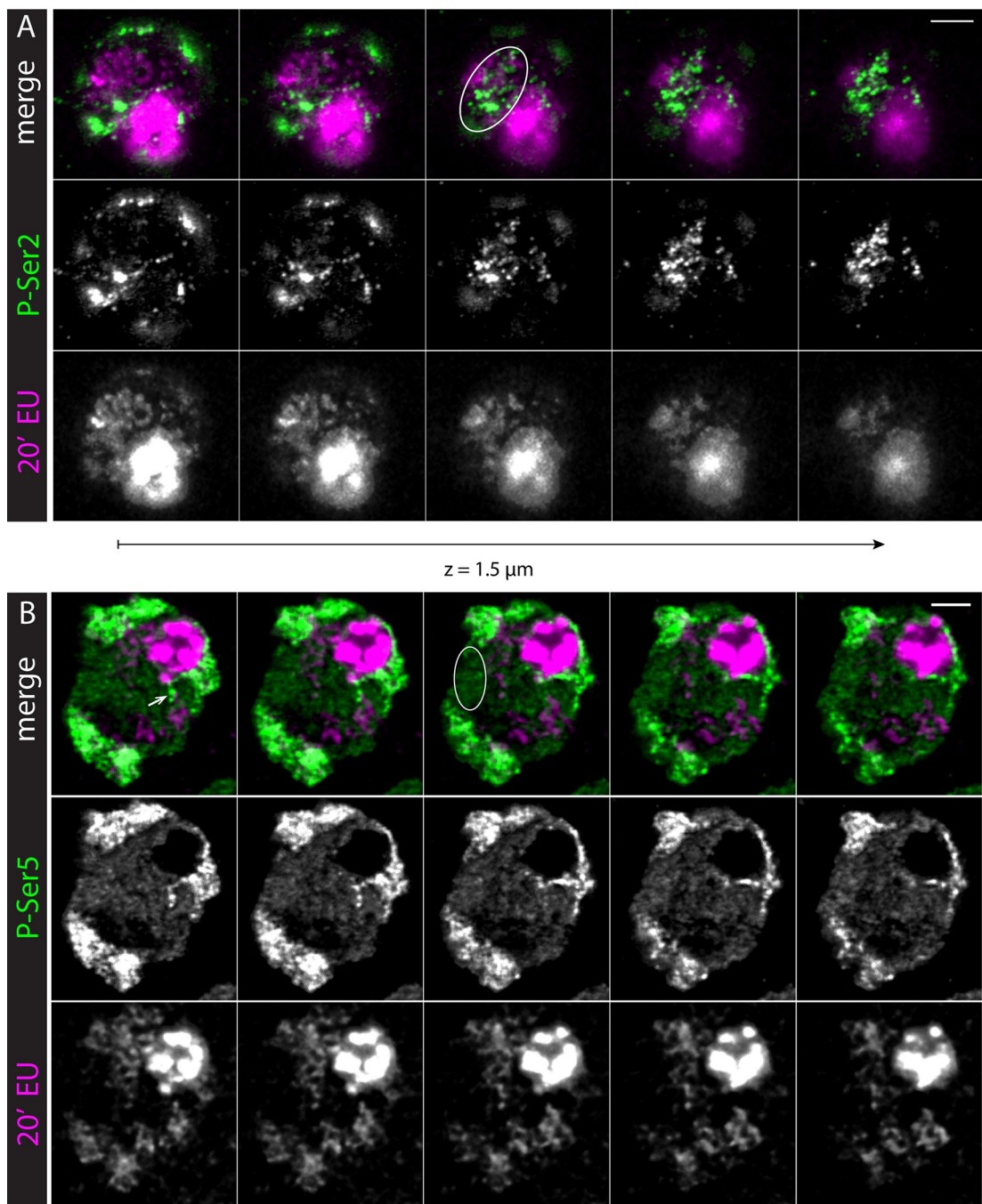

**Fig 11. Differential association of nascent RNA with RPol-PSer2 and RPol-PSer5.** Primary spermatocytes in larval testes stained for EU to label nascent RNA (bottom rows, magenta in merge) after a 20min incubation period and by immunolabelling for RPol-PSer2 (A) or RPol-PSer5 (B) (middle rows, green in merge). Consecutive confocal sections are shown from left to right covering approx. 1.5 μm along the z-axis. Scale bars, 3 μm. Nascent RNA accumulates around RPol-PSer2 along Y loops (ellipse in A), while RPol-PSer5 is frequently found without associated nascent RNA in the nucleoplasm (ellipse in B). Arrow in B points to prominent Y loop structure at the nucleolus, which is enriched with RPol-PSer5. Additionally, nascent RNA is enriched in chromosome masses in the nuclear periphery, which contain RPol with both CTD modifications, and in the nucleolus, where transcription is mostly performed by RPol1 (saturated round structure). Due to the high range of intensities some structures are saturated to visualize signals along the less dense Y loops.

of 35%) derived from EM analysis of nucleosome density in intact nuclei. In addition to the clusters, in the STORM images we also see sparse labelling between clusters indicating that the clusters are linked by nucleosomal chromatin and we also see localisations extending from the clusters supporting the occurrence of small chromatin loops emanating from the compact clusters. Our interpretation of the overall structure is illustrated in the schematic in Fig 4.

To investigate the mechanism of cluster formation we asked whether the cluster linking regions might be sites of elongating polymerase as we considered that chromatin disruption or polymerase-dependent supercoiling might underpin the chromatin cluster chain organisation. However, elongating polymerase recognised by phosphorylation on Ser2 in the CTD repeats was too sparsely distributed on the loops to account for the chromatin cluster organisation. Even though the modified polymerase form associated with paused polymerase (RPol-PSer5) was more abundant than RPol-PSer2, neither its distribution nor the combined distributions of RPol-PSer2 and RPol-PSer2 indicated a fixed relationship between active polymerase and chromatin cluster formation. Thus our evidence does not support the idea that the cluster chain organisation is generated by transcription in the intervening regions. A similar conclusion was reached by Castells-Garcia et al. [36] who suggested the independence of chromatin clutch formation and transcription. The cluster formation may represent an inherent self-aggregating property of chromatin as seen in vitro [37].

Examination of elongating polymerase (RPol-PSer2) in association with the chromatin fibre of the Y loops revealed an interesting topological relationship. The RPol-PSer2 is not simply associated with the central axis of the chromatin fibre but often is located off the fibre axis, on the periphery of a chromatin cluster (Fig 5 and schematic in Fig 6). In some cases the RPol-PSer2 is associated with sparse histone localisations extending from the chromatin clusters suggesting that the elongating polymerase is on a small chromatin loop. This arrangement of the elongating polymerase on the periphery of chromatin clusters fits with observations on mammalian cells [36], and may be relevant for the long-standing conundrum of how the polymerase transcribes double helical DNA without the transcript becoming entangled in the DNA. An attractive solution to this problem is that the polymerase is restrained from rotation, allowing the DNA to be reeled through the polymerase and so the transcript produced by this stable polymerase is not wound round the DNA [38,39]. This then poses the question of how the polymerase might be stabilised. One potential solution is that the polymerases are aggregated into large transcription factories that provide the necessary structural restraint [40]. Although we see occasional polymerase aggregates on the Y loops, in general the polymerase localisations are distributed along the loops and are not aggregated into large factories and this fits with the observation of nascent transcripts distributed in a series of blobs along the length of the Y loops (Fig 10). A similar distribution of nascent transcripts along chromatin loops is seen in long mammalian genes [41]. The localisation of the elongating polymerase in a complex on the periphery of chromatin clusters could potentially provide stability and this, together with the association of a large mass of transcript RNP from these huge transcription units [42] may provide sufficient restraint from rotation.

The spermatocyte Y loop system also enabled us to examine the progression of the two phosphorylated forms of RNA polymerase (RPol-PSer2 and RPol-PSer5) on transcription units in intact nuclei. RPol-PSer2 is distributed over the Y loops and is associated with nascent transcript as expected for active elongating polymerase. RPol-PSer5 on the other hand is concentrated on the periphery of the nucleolus presumably where Y loop transcription is initiated, however it is also seen extending a short way out from the nucleolus (Fig 11) in a position consistent with association with the beginning of Y loop transcription. This suggests that the PSer5 modification is transiently retained on the elongating polymerase as it moves away from the site of initiation. We also find weak RPol-PSer5 signal widely dispersed in the nucleoplasm

and in general this is not associated with active transcription; i.e. it is not associated with significant EU-Click-iT labelling in a 20min time window. Some of this nucleoplasmic RPol-PSer5 is associated with Y loop chromatin fibres where it may indicate paused, stalled or slowed polymerase but much is also apparently distant from chromatin and the relevance of this non-chromatin-associated RPol-PSer5 is unclear. The strong RPol-PSer5 that extends a short way out from the nucleolus has the further implication that the site of initiation is positioned close to one end of the loop emerging from the Y chromosome located close to the nucleolus. The loops may thus be co-extensive with the transcription units which would be consistent with a role for transcription in the process of loop extension.

## Materials and methods

### Antibodies

Primary antibodies were: mouse anti-histone (core histones + H1, MabE71, Millipore), 1:1000; rabbit anti-RPol-PSer2 (ab238146, Abcam), 1:500; rabbit anti-RPol-PSer5 (ab76292, Abcam), 1:500 and rabbit anti-RPol-PSer2/5 (47355, Cell Signaling); 1:500. Secondary antibodies were from Invitrogen and used at 1:1000; goat anti-mouse Ig-Alexa Fluor 647 (A-21235), goat anti-rabbit Ig Alexa Fluor 488 (A-11008) and goat anti-rabbit Ig Alexa Fluor 568 (A-11011).

### Spermatocyte Immunolabelling for STORM imaging

Testes of 0–5 day old male w1118 *Drosophila melanogaster* were dissected in PBS. The primary spermatocytes were isolated via gentle pipetting following collagenase digestion (Sigma-Aldrich C8051, 5 mg/ml in PBS for 5min at room temperature) of the testes sheath, and fixed in 4% formaldehyde/PBS for 20min at 37˚C. The primary spermatocytes were filtered (Partec 04-004-2327) and seeded onto 35 mm high μ-dishes (Ibidi) for 30min at room temperature. The cells were blocked overnight (1% Roche Western Blotting Reagent (WBR), Merck; 0.5% Triton X-100 in PBS). Following immunolabelling, the cells were fixed (4% formaldehyde) for 20min at room temperature and stored in PBS at 4˚C. To control for under-labelling of structures, we tested a range of antibody concentrations and found no significant difference in the observed nucleosome or RPol distribution.

### STORM imaging

Cells were imaged using a Zeiss Elyra 7, at 30˚C using a BP490-560/LP640 filter. The STORM imaging buffer was as described in Peters et al., 2018 [43], adapted from [44,45]. For dual-labelled samples with both Alexa Fluor 647 and 568, the cameras were aligned before each session using beads, or co-labelled structures on the cell dish (see S1 Information). Varying numbers of frames (between 20,000 and 50,000 frames) were taken to optimally capture the signal for the biological structure imaged. There was no observed difference between quantification of the Y loops across different frame numbers above 20,000 frames therefore we concluded that the chromatin structure was saturated beyond this point. The exposure time used was 30 ms, using both the 561, and 640 laser at ~ 10 kW/cm$^2$ laser power throughout. The 488 laser was used at increasing power throughout image acquisition to recover fluorophores from the triple dark state, using ~ 70–100 W/cm$^2$ power. We found the 488nm laser performed better on our microscope system as it gave a higher signal to noise than the commonly used 405nm activatory laser.

## STORM buffers

Pyranose Oxidase Enzyme Storage buffer: PIPES (Sigma P-6757) 24 mM, MgCl$_2$ (Sigma M8266) 4 mM, EGTA (Bioserv BS-7249E) 2 mM, Glycerol (Sigma G5516) 30% v/v in dH$_2$O (pH 8.4). 250U Pyranose Oxidase enzyme (Sigma P4234) was added to 1.25 ml of enzyme storage buffer, mixed well then aliquoted into 75 μl tubes for storage at -20˚C.

Pyranose Oxidase Imaging Base buffer: Tris buffer pH 8.5 (Jena Bioscience BU-124L-84) 50 mM, NaCl (Fisher Scientific 7647) 10 mM, Glucose (Sigma G8270) 0.56 M in dH$_2$O.

Pyranose Oxidase Final Imaging buffer: to 2784 μl Imaging Base Buffer was added 75 μl Pyranose Oxidase in Enzyme Storage Buffer (5 U/ml), 150 μl MEA (Sigma 30070) 1M (50 mM), 6 μl Catalase (Sigma C30)(40 μg/ml) and 30 μl COT (Sigma 138924) 200 mM in DMSO (2 mM).

## Image analysis

Resulting.czi files were processed into localisation files using Zeiss ZEN Black software. The reconstructions were filtered based on precision (>5, <40nm), photon number (>150, <8000), and point spread function (PSF) width (80 – 220nm) to improve the accuracy of the images and filter out noise. Drift correction was completed using the model-based drift correction, allowing an automatic selection of references points, a maximum of 21. The images were visualised using the "molecule density" option, which displays the localisations according to the density of neighbouring points, as well as precision value. The final images were then converted into.tif files and imported into ImageJ for the addition of scale bars. The FIJI plug-in Template Matching was used to correct any remaining misalignment of stack images (see S1 Information for details).

## Cluster analysis

The STORM-ready fluorophores Alexa Fluor 568 and Alexa Fluor 647, conjugated to the secondary antibodies used in this study blink throughout time when exposed to STORM buffer. The centres of these blinks were statistically recognised using Zen Black software SMLM processing. These central points, hereafter referred to as 'localisations' were then reconstructed into a final image. Localisations from multiple antibodies on labelled structures form clusters. To model the spatial aggregation of antibody labelled molecules, a protocol based on spatial statistics and clustering approaches was designed and implemented. First, a description of the average cluster was established using spatial statistics, then a clustering algorithm using the average cluster parameters provided by the initial spatial statistics further refined the description; finding the cluster centres and the deviation of clusters from the average model.

We hypothesized that single molecule localisations form clusters that can be described as a modified Thomas process [46]. This model is similar to a Bayesian approach [27], where the origins ("parent points") of the clusters were assumed to be completely randomly distributed, and the localisations can be seen as samples from identical isotropic 2D normal distributions N(oi,σ) around the origins oi. The modified Thomas process has a closed form pair correlation function (PCF):

$$g(r) = 1 + \frac{1}{K}\frac{1}{(4\pi\sigma_1^2)}exp\left(-\frac{r^2}{4\sigma_1^2}\right) \tag{1}$$

where $\sigma_1$ characterises the size of the cluster and $\kappa$ is the intensity of the parent process. The parameters $\sigma_1,\kappa$ can be estimated from fitting Eq 1 to the empirical PCF obtained from the data. This approach resulted in a description of the average cluster.

To mitigate the particular issue of cluster clumping caused by histones following along a Y loop fibre (violating the assumption of complete spatial randomness for the parent cluster location), a double cluster model was considered: the origin points are not randomly distributed but form a modified Thomas process as well. The PCF in this case became:

$$g_2(r) = 1 + \frac{1}{K\mu}\frac{1}{(4\pi\sigma_1^2)}exp\left(-\frac{r^2}{4\sigma_1^2}\right) + \frac{1}{K}\frac{1}{(4\pi(\sigma_2^2 + \sigma_1^2))}exp\left(\frac{-r^2}{4(\sigma_2^2 + \sigma_1^2)}\right) \qquad (2)$$

with $\sigma_1$ describing the parent cluster radius, $\sigma_2$ the scale of localisation clusters and $\mu$ the number of fluorophores per cluster. Goodness of fit was evaluated for each data selection, and the σ resulting from the best fitting cluster model (single-cluster, or double-cluster) was recorded for downstream cluster analysis. In order to identify and describe individual clusters, a mode-finding clustering algorithm, MeanShift was applied [28,29] using a MATLAB script based on [30].

The crucial parameter of MeanShift is the search radius, which was set to 2σ, as suggested in [47] and calculated for each individual dataset from the spatial statistics process. The other parameters used for MeanShift were as follows; weight 0.2, maximum iteration 50, minimum cluster size 15. The number of localisations and mean full width half maximum (FWHM) per identified clusters were recorded (the mean FWHM was considered in order to account for potential anisotropy of the clusters) and visualised using histogram and box and whisker plots [48]. The quantification workflow can be summarised as follows: 1) A section of confident non-overlapping Y loop clusters was cropped into regions of interest (ROIs), as shown in Fig 3B; 2) The 2σ of the average cluster model for each individual cropped dataset was quantified using spatial statistics to best set up MeanShift for robust clustering; 3) MeanShift was run on each dataset with the matching 2σ search radius parameter, and each cluster within each dataset was assigned a mean FWHM value; 4) all of the results were pooled and displayed as histograms and box and whisker plots (Total experiments = 7, total cells = 12, total Y loop clusters quantified = 2473).

## Nucleosome per cluster calculation

The volume of a single nucleosome was calculated as a disc of radius 5.5nm, and height 5nm, giving a nucleosome volume of 475.2nm$^3$. The maximum number of nucleosomes within a given sphere volume for each cluster was then calculated by dividing the cluster volume by the nucleosome volume.

## Transcription inhibition

20 testes per condition were dissected from male w1118 *Drosophila melanogaster* larvae, and placed into either the transcription inhibition medium consisting of 200 μg/ml α-amanitin (Sigma), 1 μM triptolide (ApexBio A3891) in Schneider's medium (Sigma) + 10% FBS (Biosera), or control medium containing 0.01% DMSO. After 19h 15min incubation in a humidity chamber at 25˚C, EU was added to 5 testes per condition. After 20h of total incubation time, testes were either processed for EU-Click-chemistry and confocal imaging of whole-mount testes or for STORM imaging of single spermatocytes.

## EU labelling and immunofluorescence of larval *Drosophila* testes

*Drosophila melanogaster* (w1118) larval testes were dissected into ice-cold Schneider's medium (Gibco) supplemented with 10% FBS (Biosera) and 1x Penicillin/Streptomycin (Sigma). For EU labelling and click chemistry the Click-iT RNA Imaging Kit (Molecular Probes, Inc.,

C10330) was used according to the manufacturer's protocol with minor modifications. Briefly, testes were incubated in medium containing 1 mM 5-ethynyl uridine (EU) for 20min (45min for transcription inhibition experiment) at 25˚C and fixed with 3.7% Formaldehyde (Sigma), 1% Triton X-100 (Sigma) in PBS (Oxoid). For detection, the incorporated EU was ligated to Alexa Fluor 647 azide via the click reaction. For immunofluorescence, testes were blocked and permeabilized with 1% Western Blocking Reagent (WBR, Merck), 0.5% Triton X-100 in PBS at 4–8˚C over night, incubated with primary antibodies in PBS containing 1% WBR, washed with 0.1% Tween 20 (Sigma-Aldrich) in PBS, incubated with secondary antibodies, washed with 0.1% Tween 20 in PBS, post-fixed for 20min with 3.7% Formaldehyde at room temperature and mounted on microscopy slides with AF1 (Citifluor).

Imaging was performed on a Leica TCS SP8 confocal microscope with an HC PL APO CS2 63x/1.40 OIL objective, 1 AU pinhole size, voxel size of 42x42x140 or 71x71x299nm, 4x line averaging using 488 Argon and 633nm HeNe lasers, and GaAsP Hybrid Detectors (HyD). The Alexa Fluor 488 channel was acquired sequentially with the 647 channel. Data were denoised with Noise2Void (3D) [49] using ZeroCostDL4Mic [50]; a comparison with the raw data is shown in S1 Information. For N2V3D model training 5-slice stacks from the bottom, centre and top of the unprocessed stack were used and training quality was assessed by comparing training and validation loss, and by visual examination of raw and denoised images. Figures were arranged and contrast adjusted with Fiji [51], Adobe Illustrator and Photoshop.

## Supporting information

**S1 Information. The anatomy of transcriptionally active chromatin loops in *Drosophila* primary spermatocytes using super-resolution microscopy: Supplementary data.** (DOCX)

**S1 Fig. Camera alignment and denoise procedure.** (TIF)

**S1 Data. Numerical data for the cluster analysis.** (CSV)

## Acknowledgments

We thank Sherif El Sharnouby and Levente Kovacs for help with spermatocyte labelling methods, Ruby Peters for help with dual label STORM and Martin Lenz for expert microscopy support.

## Author Contributions

**Conceptualization:** Robert White.

**Formal analysis:** Madeleine L. Ball, Stefan A. Koestler, Leila Muresan.

**Funding acquisition:** Robert White.

**Investigation:** Madeleine L. Ball, Stefan A. Koestler, Sohaib Abdul Rehman.

**Methodology:** Madeleine L. Ball, Stefan A. Koestler, Sohaib Abdul Rehman.

**Project administration:** Robert White.

**Supervision:** Kevin O'Holleran, Robert White.

**Writing – original draft:** Madeleine L. Ball, Stefan A. Koestler, Leila Muresan, Robert White.

**Writing – review & editing:** Madeleine L. Ball, Stefan A. Koestler, Leila Muresan, Sohaib
Abdul Rehman, Kevin O'Holleran, Robert White.

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
