## [Editor Report · Decision Letter 0]

29 Oct 2022

Dear Dr White,

Thank you very much for submitting your Research Article entitled 'The anatomy of transcriptionally active chromatin loops in Drosophila primary spermatocytes using super-resolution microscopy' to PLOS Genetics, through Review Commons.

The manuscript was fully evaluated at the editorial level, taking into account the Review Commons peer review reports and your Revision Plan.  

We will be happy to consider a revised manuscript, and we generally agree with the revision plan that you propose.  We cannot, of course, promise publication at that time. We do concur with reviewer #2 that the data in figures 9 and 10 need to be generated by STORM rather than by conventional confocal microscopy. Although not explicitly mentioned by the reviewers, it also seems to us that the conclusions drawn from these experiments need to be backed up by quantitative image analysis, rather than by anecdotal images alone. Additional experiments that study the effect of transcription inhibition on the cluster organisation (as suggested by Reviewer #1) would also strengthen the manuscript significantly. Minor point: in the legend of Fig 3B, please explain what the three colors are.

Should you decide to revise the manuscript for further consideration here, please update the list of your responses to the review comments and a description of the changes you have made in the manuscript.  If you decide to revise the manuscript for further consideration at PLOS Genetics, please aim to resubmit within the next 60 days, unless it will take extra time to address the concerns of the reviewers, in which case we would appreciate an expected resubmission date by email to plosgenetics@plos.org.

You can use this link to log into the system when you are ready to submit a revised version, having first consulted our Submission Checklist.

Please do not hesitate to contact us if you have any concerns or questions.

Yours sincerely,

Bas van Steensel

Academic Editor

PLOS Genetics

Gregory P. Copenhaver

Editor-in-Chief

PLOS Genetics

---

## [Decision Letter · Decision Letter 1]

4 Feb 2023

Dear Dr White,

We are pleased to inform you that your manuscript entitled "The anatomy of transcriptionally active chromatin loops in Drosophila primary spermatocytes using super-resolution microscopy" has been editorially accepted for publication in PLOS Genetics. Congratulations!

Please also check and adhere to the Data Availability policy of PLOS Genetics, as outlined below. 

Yours sincerely,

Bas van Steensel

Academic Editor

PLOS Genetics

Gregory P. Copenhaver

Editor-in-Chief

PLOS Genetics

Comments from the reviewers (if applicable):

Reviewer's Responses to Questions

**Comments to the Authors:**

Reviewer #1: The authors have fully addressed my comments. This work is of high interest and perform in a robust manner. Overall, this manuscript will be of high interest to the community.

Reviewer #2: The authors addressed the majority of the comments satisfactorily. Some of the suggestions were not addressed probably after an agreement with the editor. I would like just to make some minor remarks.

In response to this comment “Given the availability of broadly used methods for clustering such as DBScan …The authors could justify the choice or compare results using other more broadly used methods.”

The authors answered “…In selecting this approach we compared various approaches and we found Meanshift to perform best with respect to identifying clusters closely positioned along a fibre. (Example data comparing the performance of Meanshift with DBscan in cluster identification with varying radius input parameter was provided to reviewers).”

I would like to mention that I was not provided any example data.

In the response to Reviewer #1 – Comment 1, the authors stated “we tried a range of antibody concentrations which did not change the observations” referring to labeling density of nucleosomes and RNA Pol II. Since they included a related statement in the Methods section it would be appropriate to include supplementary panels supporting the statement or at least show the data to the reviewers.

**Have all data underlying the figures and results presented in the manuscript been provided?**

Reviewer #1: Yes

Reviewer #2: None

PLOS authors have the option to publish the peer review history of their article (what does this mean?). If published, this will include your full peer review and any attached files.

Reviewer #1: No

Reviewer #2: No

**Data Deposition**

http://datadryad.org/submit?journalID=pgenetics&manu=PGENETICS-D-22-01212R1

**Press Queries**

---

## [Editor Report · Acceptance letter]

21 Feb 2023

PGENETICS-D-22-01212R1 

The anatomy of transcriptionally active chromatin loops in Drosophila primary spermatocytes using super-resolution microscopy 

Dear Dr White, 

We are pleased to inform you that your manuscript entitled "The anatomy of transcriptionally active chromatin loops in Drosophila primary spermatocytes using super-resolution microscopy" has been formally accepted for publication in PLOS Genetics! Your manuscript is now with our production department and you will be notified of the publication date in due course.

With kind regards,

Judit Kozma

PLOS Genetics

On behalf of:
